An efficient approach for low latency processing in stream data

http://orcid.org/0000-0002-8105-3563 Bhatt Nirav 1 niravbhatt.it@charusat.ac.in
Thakkar Amit 2
1 Information Technology, Chandubhai S Patel Institute of Technology, CHARUSAT , Anand, Gujarat , India
2 Computer Science and Engineering, Chandubhai S Patel Institute of Technology, CHARUSAT , Anand, Gujarat , India
Soufan Othman
Electronic publication date: 2021 Mar 10
Publication date: 2021
Volume: 7
Electronic Location ID: e426
Received 2020 Nov 9; Accepted 2021 Feb 12
Copyright: © 2021 Bhatt and Thakkar
Copyright year: 2021
Copyright holder: Bhatt and Thakkar
License: This is an open access article distributed under the terms of the Creative Commons Attribution License, which permits unrestricted use, distribution, reproduction and adaptation in any medium and for any purpose provided that it is properly attributed. For attribution, the original author(s), title, publication source (PeerJ Computer Science) and either DOI or URL of the article must be cited.
License URL: https://creativecommons.org/licenses/by/4.0/

Keywords: Data stream, Stream processing, Latency

Funding: The authors received no funding for this work.

==============================
Stream data is the data that is generated continuously from the different data sources and ideally defined as the data that has no discrete beginning or end. Processing the stream data is a part of big data analytics that aims at querying the continuously arriving data and extracting meaningful information from the stream. Although earlier processing of such stream was using batch analytics, nowadays there are applications like the stock market, patient monitoring, and traffic analysis which can cause a drastic difference in processing, if the output is generated in levels of hours and minutes. The primary goal of any real-time stream processing system is to process the stream data as soon as it arrives. Correspondingly, analytics of the stream data also needs consideration of surrounding dependent data. For example, stock market analytics results are often useless if we do not consider their associated or dependent parameters which affect the result. In a real-world application, these dependent stream data usually arrive from the distributed environment. Hence, the stream processing system has to be designed, which can deal with the delay in the arrival of such data from distributed sources. We have designed the stream processing model which can deal with all the possible latency and provide an end-to-end low latency system. We have performed the stock market prediction by considering affecting parameters, such as USD, OIL Price, and Gold Price with an equal arrival rate. We have calculated the Normalized Root Mean Square Error (NRMSE) which simplifies the comparison among models with different scales. A comparative analysis of the experiment presented in the report shows a significant improvement in the result when considering the affecting parameters. In this work, we have used the statistical approach to forecast the probability of possible data latency arrives from distributed sources. Moreover, we have performed preprocessing of stream data to ensure at-least-once delivery semantics. In the direction towards providing low latency in processing, we have also implemented exactly-once processing semantics. Extensive experiments have been performed with varying sizes of the window and data arrival rate. We have concluded that system latency can be reduced when the window size is equal to the data arrival rate.

Introduction

Due to the growth of social media and its applications, which demand a massive amount of data, the data generation rate is high and continuous. They are known as stream data. It is not feasible to store such data substantially, and usual data mining methods cannot handle it. So we need a system or model to analyze and manage such stream data, which is called a stream management system (Bhatt & Thakkar, 2019a).

Commonly used terms which are related to stream data are,Unbounded data: The data which is not fixed like batch data, and the data which is continuous and endless is called unbounded data. There is a need to have a specific type of execution engine for processing streaming data (Akidau, Chernyak & Lax, 2018; Akidau, 2015)

Unbounded data processing: A continuous processing of unbounded data (Akidau, 2015). Batch processing becomes a subset of stream processing (Bhatt & Thakkar, 2019a).

Low-latency, approximate, and abstract results: Unlike batch processing systems, streaming engines are related to low-latency or hypothetical (Akidau, Chernyak & Lax, 2018; Akidau, 2015).

As stream management models cannot observe and compute the entire data exactly, some kind of approximation is required (Masseglia et al., 2008). Therefore, processing of stream data is also crucial at the same time complex, and the causes are (Bhatt & Thakkar, 2019a):Businesses process the data having a particular timestamp and need to process in time order. Optimizing stream processing is a better way to deal with the issue of latency.

Large and infinite data that are in recent trade are efficiently controlled with a specific system that is intended for such continuously arriving data.

Processing data in such a system is a challenging task and requires an effective processing mechanism that is well defined by considering the processing of continuously arriving data.

Stream data are often assumed to arrive from a variety of heterogeneous data sources, at high velocities, and in large volumes. How to retrieve valuable information based on the high-speed stream data and massive historical data has become a new challenge in the field of data mining to perform analysis immediately on incoming data (Wu, 2014; Akidau et al., 2015). The essential task of any streaming system is, processing arriving data from scattered sources and generate an output promptly. The critical deliberations for that desired task are latency and throughput. Hence dealing with stream imperfections such as late data, lost data and out-of-order data becomes significant research in big data stream processing (Bhatt & Thakkar, 2019b).

There are many real-world stream data based applications where the prediction of such an application depends on the value of different distributed data sources as well. For example, to perform the prediction on the stock market data, we also need to consider the value of the other relevant parameters which affects the stock market, as there exist dependency between the price of gold, oil, USD and the stock market (Arfaoui & Ben Rejeb, 2017; Bedoui et al., 2018). Hence any delay in incoming data from different distributed sources will affect the prediction. So there is a need that one should forecast the probability of delay in the incoming streams of data through the appropriate statistical method.

Many challenges associated with stream data processing: (1) Since the source of continuously arriving stream data are distributed and their processing is dependent on other distributed parameters too, delay in any parameter introduces different types of latency such as data latency, system latency, and processing latency. (2) Statistical forecasting of the latency in the stream processing system to define the allowed lateness for a more accurate stream processing result. (3) To handle the repetitive processing of similar incoming data which will eventually increase the latency in overall stream processing. Henceforth, there is a need to identify all the different types of latency present in the stream processing system. This paper presents the work to perform effective stream data processing with possible low latency.

The main contributions of our work are summarized as follow:We discover the strong relationship between the movement of the stock market and their dependent parameter. Accordingly, we design and implement the stream processing pipeline, which can process the distributed and continuously arriving stream data along with their dependent parameters with an equal arrival rate.

Normalized Root Mean Square Error (NRMSE) is calculated to measure the effect after considering the co-movement of related parameters for stock market prediction.

We explore the different types of latency, such as data latency and system latency in the stream processing system. Accordingly, we design and implement the proposed, end-to-end low latency, stream processing model which deals with data latency and system latency.

The statistical forecasting of the data latency in the stream processing system:

Compare and identify the appropriate latency distribution model.

Forecast the data latency through appropriate probability density function and hazard rate.

The proposed system has been implemented by considering the “Exactly-Once-Delivery” and “Exactly-Once-Processing” semantics to ensure the low latency in stream processing. Finally, an experimental comparison between different window sizes and data arrival rate is presented.

Literature survey

A variety of approaches have been proposed to provide effective stream processing. Earlier stream data processing is used to be done by using the series of batch processing internally. Processing the stream data in small batches is still a time-consuming process which adds many types of delay in a system. Although, researchers have started working on batched stream processing systems that provide higher throughput with considerably lower latency. The work presented in Zaharia et al. (2013) and Wu et al. (2017) defines the working of the batched stream processing system. However, results show such systems do not fit for stream fluctuation and uneven workload distribution. Semeniuta & Falkman (2019) and Gabriel et al. (2020) state that periodic sampling became a dominant strategy for real-time stream processing systems. Although the periodic approach works well, in the distributed environment, one faces such challenges as latency. Carbone et al. (2015) has presented processing of batch and stream in a single-engine where they analyzed, processing that is based on event-time may exhibit latency due to event-time processing-time delay. Affetti et al. (2017) have analyzed the different stream processing window systems such as time window and count window to deal with latency. The choice of an appropriate window mechanism for stream processing is based on application. Although, under processing time semantics, the processing speed of the machines, as well as temporary overloads, might impact the output produced. Miao et al. (2017) have presented the work on stream processing in a multi-core machine. They have presented a novel stream processing engine called “StreamBox”, where the primary design goal is to minimize latency with the help of epoch and pipeline parallelism. As the consideration of stream processing semantics can also affect the latency, the choice of appropriate semantic may lead to the lowest latency. Event-time latency and processing-time latency have been explored in the recent work presented in Karimov et al. (2018) to provide possibly low latency stream processing. On the other hand, there is another possibility of latency into the stream processing system due to delay in arriving stream data, known as data latency. The state in the stream processing system, where data might have arrived into the system, but awaiting the processing also introduces the latency, known as the system latency. Akidau et al. (2015) have defined the data flow model for balancing correctness and latency in processing out-of-order data. The author has used the concept of a watermark, which is a belief of input completeness concerning event times, to distinguish the data delay from the system delay. As the possibilities of drifts in stream data is a common issue, and to address the duration- based failure, the work presented in Dong & Cui (2021) has considered two-stage degradation in a dynamic environment specifically when the threshold is a random variable. Similar work to handle the delay through the threshold, (Huang et al., 2021), has proposed a time-varying threshold for event-triggered mechanism using backstepping design. Authors in Chen et al. (2020) have presented fuzzy adaptive two-bits-triggered control specifically for the uncertain system. They have concluded that addressing the input saturation for continuously arriving data and studying the inter-execution interval is important for an effective triggering mechanism. To provide low-latency message forwarding in sparse setting, a new opportunistic network framework called WON has been proposed by Fu et al. (2019) which consider all the message-forwarding activities such as mobile-to-mobile, mobile-to-stationary, stationary-to-mobile and stationary-to-stationary. Although authors have explored that, automatically get the recommended settings according to the input requirements can be an effective future work. To handle the time-varying delay related issues in stream processing, initial work presented in Zhang et al. (2011) has analyzed Markovian jump systems. The improvement in the same was proposed by Xiong et al. (2016), where they have studied the delay-dependent stability and suggested the Lyapunov functional with the delay decomposition technique using Jensen inequality. Although authors have concluded there is scope to explore new integral inequalities to further improve the delay-dependent stabilization.

Interdependencies between stock market data with dependent parameters

In the real world, an event-time based stream processing system, more specifically, the one where the prediction depends on multiple independent parameters suffer from different types of latency and low throughput. Existing windowing and triggering approach unable to handle different types of latency and to provide higher throughput for the unbounded data.

As Arfaoui & Ben Rejeb (2017) have analyzed, there is a dependency between the value of gold, USD, OIL and the stock market. The purpose is to examine, from a global perspective, the oil, gold, USD and stock prices interdependencies and to identify instantaneously direct and indirect linkages among them (Arfaoui & Ben Rejeb, 2017). For instance, to perform stock market analysis, it is advisable to consider the co-movement, which has a high impact on the stock market analysis. Likewise, a recent study carried out by Bedoui et al. (2018) proposes a nested copula-based GARCH model to explore the dependence structure between oil, gold and USD exchange rate. More importantly, a comparative framework based on three sub-periods is implemented to capture the co-movement during the regular and crisis period. Empirical results suggest that for both crisis periods the dependence between oil, gold and USD exchange rate along with stock market value is stronger compared with the dependence during the untroubled period (Bedoui et al., 2018).

Experimental analysis of stock market data

We have performed experiments on the Apache Beam. Apache Beam is a platform that allows streaming data. Apache Beam incorporates the different functionalities provided by Google and Apache independently into a single platform. We have used “Dataflow Runner” as a Beam Runner (Google, 2021) which runs on Google Cloud Platform. The experiments have been performed on the stock market prediction application using the following Algorithm 1 of linear regression with multiple variables. The dataset considered of the BSE stock market with USD, gold price and Oil Price is described in Table 1.

Table 1 Dataset description.

Data	Source	Attributes	Duration (year)	
BSE Stock Market Data	Investing.com (2019)	Date, Open, High, Low, Close	2000–2019	
USD Price (USD)	Federal Reserve Bank of St. Louis. (2020)	USD value	
Gold Price (INR)	Goldprice (2020)	Gold value	
Crude Oil Price (INR)	Oil value	

Algorithm 1 Linear regression with multiple variable.

Step 1: Prediction	
y^(x)=θ0+θ1x1+θ2x2+θ3x3=θxT, where x1=USD,x2=Gold_Price,x3=Oil_Price	
Step 2: Cost function	
J(θ)=12m∑j⁡(y(j)−y^x(j)T)2	
J(θ)=12m∑j⁡(yT−θxT)(yT−θxT)T	
Step 3: Optimization of hyperparameter by stochastic gradient descent algorithm	
∇J(j)(θ)=−2m∑j⁡(y(j)−θx(j)T).[x0(j)x1(j)…]	

We have performed experiments on the Apache Beam. Apache Beam is a platform that allows streaming data. Apache Beam incorporates the different functionalities provided by Google and Apache independently into a single platform. We have used “Dataflow Runner” as a Beam Runner (Google, 2021) which runs on Google Cloud Platform. Experiments have been carried out on the stock market prediction application using the following Algorithm 1 of linear regression with multiple variables.

Step-1 in Algorithm 1 specifies the prediction function y^(x). We have considered the additional parameter like USD, Gold Price and Oil Price. Step-2 shows the calculation of the cost function by considering the difference between the actual (y) and predicted (y^) value. We have optimized the hyperparameter using stochastic gradient descent, as shown in Step-3. Figure 1 shows the comparative analysis on the actual close value of BSE Stock market data with prediction considering dependent parameters. We have also provided comparative results between prediction with considering parameters and without considering parameters in Fig. 2. We have calculated the NRMSE, as shown in Eq. (1), which is the RMSE facilitates the comparison between models with different scales.

Figure 1 Actual vs. prediction considering dependent parameters.

Figure 2 Comparative analysis on prediction of stock market data with dependent parameters and without dependent parameters.

(1) NRMSE=RMSEO¯

where, O¯ in Eq. (1) is the average of observation value and RMSE can be calculated as shown in Eq. (2),

(2) RMSE=100∗∑i=1n⁡(Xobs,i−Xmodel,i)2n

where, X (Obs, i) in Eq. (2) is the observation value, and X (model, i) is the forecast value. Generally, RMSE is preferred when we are comparing different models on the same data while NRMSE is the statistical measure that is suitable for comparing different data scales. Hence, the NRMSE can be interpreted as a fraction of the overall range that is typically resolved by the model. As shown in Fig. 3, we have observed the NRMSE for the prediction considering dependent parameters is 0.03 (3%) which is significantly better than the NRMSE of 0.09 (9%) for the prediction without dependent parameter.

Figure 3 NRMSE for observed data.

Finding from the experimental survey is, in a distributed environment, any parameter is missing or having delay/latency in processing affect the prediction. As the prediction in such stream data-based application has a dependency on co-movement of different related parameters, there is a need to design a model that captures all the possible latency and also handle the late data with the appropriate statistical method to get an accurate result.

Materials and Methods

Proposed system

In this section, we have presented the proposed pipeline, as shown in Fig. 4, followed by the proposed model for stream processing, as shown in Fig. 5.

Figure 4 Proposed pipeline.

Figure 5 Proposed model for stream processing.

The proposed pipeline presented in Fig. 4 has been implemented with reading, Predict and Write transform and executed on google cloud dataflow as a runner in apache beam. The pipeline represents two things PCollection and PTransform. PCollection accepts the price of stock, gold, oil and USD as input from distributed sources, and converted into distinct PCollection objects on which PTransform can operate. PTransform presents the processing function which can be performed on an element of the PCollection provided.

The proposed model shown in Fig. 5 accepts the continuously arriving stream data from the distributed environment and passes it on to the Google Cloud Pub/Sub. Pub/Sub is a Publish/Subscribe model of distributed systems over Google cloud. The service of pub/sub is one that is based on the publish/subscribe method and make it possible through push and pull messages. It can take data from any source or any part of the world and provide for further processing. The proposed pipeline, incorporated into the proposed model, will be executed on a dataflow runner with different transformations such as read, extract, predict, format, and write. The output of the dataflow runner will be stored back to the bucket created on the cloud storage.

The mathematical formulation of the proposed model

The proposed mathematical model for low latency stream processing is defined as follow:

Results and discussion

We have implemented the same instance rule as defined in Algorithm 2, which is the appropriate DQR for the input stream, to ensure the exactly-once delivery semantic in stream processing. Furthermore, to handle the possible data latency, we have followed the process defined in Fig. 6. We have performed event history analysis as defined in Algorithm 3, which is the process to analyze the behaviour of the continuously arriving incoming stream data. As defined in step-1 of Algorithm 3, to find the best suitable distribution model for our input data, we have identified different data distribution models that are applicable to continuously arriving stream data such as gamma distribution, Weibull distribution, and lognormal distribution. Figure 7 shows the goodness of fit by maximum likelihood estimation for the above-mentioned, statistical data distribution models.

Figure 6 Process to identify data latency.

Figure 7 The goodness of fit by maximum likelihood estimation.

(A) Histogram and theoretical densities of different data distribution model. (B) Empirical and theoretical CDFs. (C) Q-Q plot of data distribution models. (D) P-P plot of data distribution models.

Algorithm 2 Data stream preprocessing: duplicate instance rule.

Input: A sequence of input stream S = s1s2…sn.	
Output: Preprocessed stream.	
Apply the Data Quality Rule (DQR) on input data // Duplicate Instance Rule	
• Find Duplicate: Given an input stream S = s1s2…sn, where si ∈ [m] and n > m, find a ∈ [m], which appears more than once.	
         Begin	
           For each si ∈ S do	
             Get K records si+1 … sk	
             If any of the above k records are similar to si then	
               Flag = “Y”	
               Skip the input stream.	
             Else	
               Flag = “N”	
               Consider the input stream.	
           End For	
         End	

Algorithm 3 Event history analysis.

Input: A sequence of input data latency L= l1, l2 … ln.	
Distribution model M = {Gamma, LogNormal, Weibull}.	
Output: Probability density function for distribution model, Hazard rate.	
 • H(⋅|x) is a hazard rate that models the probability of an event to occur at time t.	
 • F(t) is the Probability Density Function for input data.	
 • μ = Scale, σ = Shape.	
Step-1: Identify the suitable distribution model m ∈ M. // find the suitable distribution model	
Step-2: Calculate the probability distribution function for model m. // Fit the model	
F(x)=e−((ln⁡(x/m))2/(2σ2))xσ2π	
Step-3: Calculate hazard rate H(x,σ)=(1xσ)ϕ(ln⁡xσ)Φ(ln⁡xσ) // Forecast the probability of delay	

From statistical experiments, we have found the “Lognormal Distribution” fit as the best data distribution model for our input, as shown in Fig. 8. As shown in step-2 of the Algorithm 3, we have calculated the probability density function (PDF) for “Lognormal Distribution” as follow:

Figure 8 Lognormal probability distribution.

(A) Histogram and theoretical density of lognormal probability distribution. (B) Q-Q plot of lognormal probability distribution. (C) Empirical and theoretical CDFs. (D) probability plot (P-P plot) of lognormal probability distribution.

(3) F(x)=e−((ln⁡(x/m))2/(2σ2))xσ2π

where, the value of μ,mandσ in Eq. (3) have been calculated using statistical computing tool R for Lognormal Distribution.

μ = Log(m) = −0.2359

m = Scale = 0.7898

σ = Shape = 1.8964

Hence,

(4) F(x)=e−((ln⁡(x/(0.78)))2/(2(1.89)2))x(1.89)2π

Equation (4) has been used as the probability density function of lognormal distribution to forecast future latency.

To forecast and handle the data latency, we have performed the survival analysis using hazard rate H(·|x), which models the propensity of the occurrence of an event, that is, the probability of an event to occur at time t as shown in Eq. (5).

We have defined the hazard rate to forecast the data latency as shown in Eq. (5),

Hazard rate

(5) H(x,σ)=(1xσ)ϕ(ln⁡xσ)Φ(ln⁡xσ)

where ϕ is the probability density function (PDF) of the lognormal distribution and Φ is the cumulative distribution function of the lognormal distribution. Figure 9 shows the probability of future data latency based on the hazard rate. From the experimental result, our proposed system has discovered the average data latency of 8 s, and hence after reaching the end of the window, the system will wait for the next 8 s to consider probable late data before processing.

Figure 9 Probability of future latency.

Exactly-once stream processing semantic

There is various semantics such as at-least-once, at-most-once, and exactly-once exist for stream processing. To ensure end-to-end low latency stream processing, we have implemented exactly-once stream processing semantics at our publisher-subscriber mechanism of cloud dataflow. Figure 10A shows the publisher-subscriber mechanism.

Figure 10 Exactly once processing through publisher-subscriber mechanism.

(A) Publisher-Subscriber mechanism. (B) Publisher publishes the message to the topic with unique ID. (C) Duplicate messages are identified and discarded from the topic based on unique ID.

The publisher will publish the message in topic whereas the subscriber will pull the messages from the topic. A unique data_id has been assigned to every input data; however, they are uniquely identified in the publisher-subscriber mechanism by a combination of (publisher_id + data_id), as shown in Fig. 10B. The Publisher-subscriber mechanism work based on acknowledgment. If the publisher does not receive an acknowledgment from the subscriber, the publisher may publish the same data again. To achieve exactly-once processing in our stream processing system, redundant data can be identified and discarded from the topic based on the unique (publisher_id + data_id), as shown in Fig. 10C.

To discover the low system latency, we have performed the experiments with different data arrival rate (data freshness) and different window sizes. Figure 11A shows the latency values (in seconds) for different window sizes at a data arrival rate of 15 s and Fig. 11B shows the graphical representation of the same. Likewise, Figs. 12–14 show the latency values for different window sizes at a data arrival rate of 30, 45 and 60 s, respectively. We have calculated and observed the latency values up to 10 min at an interval of every 1 min.

Figure 11 System Latency with Data Freshness 15s.

(A) System latency measures with different window size and data freshness 15s. (B) A comparative graph shows the latency values for different window size with DF = 15s.

Figure 12 System Latency with Data Freshness 30s.

(A) System latency measures with different window size and data freshness 30s. (B) A comparative graph shows the latency values for different window size with DF = 30s.

Figure 13 System Latency with Data Freshness 45s.

(A) System latency measures with different window size and data freshness 45s. (B) A comparative graph shows the latency values for different window size with DF = 45s.

Figure 14 System Latency with Data Freshness 60s.

(A) System latency measures with different window size and data freshness 60s. (B) A comparative graph shows the latency values for different window size with DF = 60s.

We have concluded from the experiments: (1) when the size of the window is larger than the rate of arrival of data, the window will wait for input data before the trigger fires which may increase the delay in the result. (2) When the size of the window is smaller than the rate of arrival of data, processing of the data may be delayed as data might have arrived into the system but waiting for processing. (3) The lowest latency can be achieved when the window size is equal to the data arrival rate.

Conclusions

To provide the low latency in stream processing, there should be a systematic flow design that can process the continuously arriving stream. We have designed the proposed pipeline for our stock market prediction application. To implement effective stream processing, one should be able to capture all the different types of delay into the system and deal with the delay through an effective statistical approach. We have designed a proposed model that provides end-to-end low latency processing. However, to provide a better prediction for continuously arriving stream data, there is a need to consider the dependent parameter from the distributed environment with an equal data arrival rate. We have performed experiments for the prediction in the stock market by considering their dependent parameter (price of oil, gold and USD), and proved that delay/absence of dependent parameters would affect the performance. Hence, we have forecasted the probability of late data through a statistical approach and implement the system which deals with such data latency accordingly. System latency can also be reduced by choosing appropriate stream processing semantics, we have implemented exactly-once stream processing semantics. We have concluded from the experiments performed on the Google Cloud dataflow that, in stream data processing, there is a dependency between the size of the window and data freshness. To reduce the overall system latency, we need to consider the appropriate size of the window as per the rate of arrival of data.

The future scope of this work is to compare the result of stream processing with other stream processing semantics such as at-least-once and at-most-once stream processing. The work can further be extended by exploring more effective data quality rules for stream data which can further reduce the delay of overall processing.

Supplemental Information

Supplemental Information 1 Dataset and code.

The cloud script and related dataset

Click here for additional data file.

Additional Information and Declarations

Competing Interests

Author Contributions

Data Availability

The authors declare that they have no competing interests.

Nirav Bhatt conceived and designed the experiments, performed the experiments, analyzed the data, performed the computation work, prepared figures and/or tables, authored or reviewed drafts of the paper, and approved the final draft.

Amit Thakkar conceived and designed the experiments, analyzed the data, authored or reviewed drafts of the paper, and approved the final draft.

The following information was supplied regarding data availability:

Source codes and data are available in the Supplemental Files.

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
