# Peer review of "An efficient approach for low latency processing in stream data"

_PeerJ Computer Science, doi:10.7717/peerj-cs.426_

## Round 0.1 · original submission · Minor Revisions

After careful consideration of reviewers' comments, I recommend a minor revision for this work. One of the reviewers suggested adding a substantial list of references. The authors are not imposed to include all of the references but to address the comment of improving the literature section of the study in a reasonable way. Still, other comments for providing more details regarding the dataset and experiments are required to be addressed.

Reviewer 1 ·

Basic reporting

The authors need to cite the following and other related newly research papers in the introduction and literature section 10.1109/MIS.2019.2942836 ; 10.1016/j.future.2018.08.031 ; 10.1016/j.ress.2020.106815 ; 10.1016/j.comnet.2020.107327 ; 10.1016/j.amc.2014.11.064 ; 10.1016/j.nahs.2015.07.005 ; 10.1155/2019/7875305 ; 10.1016/j.ins.2020.02.051 ; 10.1002/rnc.3980 ; 10.1016/j.isatra.2016.11.002 ; 10.1002/oca.2326 ; 10.1016/j.autcon.2019.02.014 ; 10.1016/j.aei.2019.100960 ; 10.1016/j.autcon.2019.102859 ; 10.1109/TIP.2018.2881828 ; 10.1016/j.amc.2015.06.036 ; 10.1016/j.autcon.2010.09.011 ; 10.1016/j.future.2020.08.021 ; 10.1109/TII.2019.2952565 ; 10.1016/j.swevo.2020.100697 ; 10.1007/s40857-020-00175-5 ; 10.1007/s11390-020-0350-4 ; 10.1109/TSP.2020.3007313 ; 10.1007/s11265-020-01610-6 ; 10.1093/imaman/dpaa009 ; 10.1109/TNNLS.2019.2955287 ; 10.3390/app10217924 ; 10.23919/JCC.2020.03.011 ; 10.1049/iet-map.2020.0090 ; 10.1631/FITEE.2000229 ; 10.1109/LED.2019.2903430 ; 10.1109/UCMMT47867.2019.9008340 ; 10.1109/LWC.2020.2982637 ; 10.1109/TCYB.2020.2970736 ; 10.1109/TITS.2020.3013928 ; 10.1109/TETC.2020.2974183 ; 10.3966/160792642020072104022 ; 10.1002/acs.3098 ; 10.1016/j.asoc.2020.106372 ; 10.1007/s12555-019-0972-x ; 10.1016/j.isatra.2020.08.022 ; 10.1504/IJDMB.2013.056078

Experimental design

no comment

Validity of the findings

no comment

Additional comments

The authors are advised to improve the research GAP. Finally, the authors are suggested to suggest such type of suggestion that are practically helpful.

·

Basic reporting

This paper presents quite a good work on stream data processing. I believe this work is interesting and in the scope of PeerJ. I am aware that latency is the biggest issue in stream processing. The statistical approach used by the author to handle latency is a good contribution. In fact, exploring different types of possible latency and defining an efficient approach to handle it is really interesting. This paper proposed a model for end-to-end low latency stream processing. The overall framework makes sense and the logical structure of the paper is very clear.
Figures show appropriate structure and are well described. The proposed approach is accepted.

Experimental design

Experimental design satisfies the relevance of the proposed work but the dataset used should be explained more and if possible in some suitable format. Author has put an effort to design a suitable dataset to explore the dependence and its effect on latency. Statistical experiments are interesting and relevant to the work.

Validity of the findings

Well organized literature findings. The objectives of manuscript is well defined and covered.

Additional comments

The paper includes good theoretical justification. The Paper is well written and strongly accepted, however some English statements can be reformulated. I also feel if future direction of this work is defined in brief to continue this research further. Overall a good piece of work.

---

## Round 0.2 · accepted · Accept

The authors have addressed the final minor comments of the reviewers.